# An Entropy-Based Approach for Anomaly Detection in Activities of Daily Living in the Presence of a Visitor

**DOI:** 10.3390/e22080845

**Published:** 2020-07-30

**Authors:** Aadel Howedi, Ahmad Lotfi, Amir Pourabdollah

**Affiliations:** School of Science and Technology, Nottingham Trent University, Clifton Lane, Nottingham NG11 8NS, UK; aadel.howedi2013@my.ntu.ac.uk (A.H.); amir.pourabdollah@ntu.ac.uk (A.P.)

**Keywords:** activity recognition, independent living, activities of daily living, anomaly detection, behavioural patterns, approximate entropy, sample entropy, fuzzy entropy, multiscale-fuzzy entropy

## Abstract

This paper presents anomaly detection in activities of daily living based on entropy measures. It is shown that the proposed approach will identify anomalies when there are visitors representing a multi-occupant environment. Residents often receive visits from family members or health care workers. Therefore, the residents’ activity is expected to be different when there is a visitor, which could be considered as an abnormal activity pattern. Identifying anomalies is essential for healthcare management, as this will enable action to avoid prospective problems early and to improve and support residents’ ability to live safely and independently in their own homes. Entropy measure analysis is an established method to detect disorder or irregularities in many applications: however, this has rarely been applied in the context of activities of daily living. An experimental evaluation is conducted to detect anomalies obtained from a real home environment. Experimental results are presented to demonstrate the effectiveness of the entropy measures employed in detecting anomalies in the resident’s activity and identifying visiting times in the same environment.

## 1. Introduction

The population of adults aged 65 or over is estimated to be more than 1.92 billion globally by the year 2050 [1]. This has a serious influence on the healthcare sector, with the cost of elderly care expected to increase enormously over the coming years [2,3]. Additionally, researchers have demonstrated that the number of older adults living alone at home and the number of single-occupancy homes are also growing worldwide, due partly to the high expense of residential care services [4,5,6]. The majority of older adults require continuous help in their Activities of Daily Living (ADL). Nevertheless, most older adults prefer to stay in their own homes for as long as possible rather than in residential or home care facilities to maintain their independence [7,8]. In order to support older adults to live independently in their own homes, the home environments equipped with appropriate sensors, referred to as Intelligent Environments (IE) or Smart Homes (SH), are used to support individuals with their daily activities, improve their quality of life, and allow them to stay safely and independently in their own homes [9,10,11,12]. To be able to support independent living for older adults, it would be essential to have a means of monitoring to recognise their daily activities and to detect anomalies in the recognised activities.

Anomaly detection, also known as abnormality detection, aims to detect and identify any abnormal patterns in activities of daily living. An unusual pattern significantly different from behavioural routine is referred to as an anomaly, and maybe an early symptom of Mild Cognitive Impairment (MCI) or of dementia in older adults [13,14]. Most of the current research in detecting an anomaly in ADLs focuses on a single-occupant environment where only one individual is monitored. The hypothesis that home environments are occupied by one resident all the time is not usually the case. It is common for the resident to receive visits from family members or health care workers. Visiting is considered as one of the most significant activities for older adults living alone at home [15]. Therefore, the resident’s activity pattern is expected to be different when there is a visitor in the same environment (represented as a multi-occupancy environment), which can also be considered as an abnormal pattern in the resident’s activities. The behaviour of a person could vary due to some personal factors such as visits and the influence of health conditions. Reliable anomaly detection in ADLs, or identifying visiting times (e.g., visits made by healthcare workers) is considered as one of the most important components of many home health care applications [5]. Thus, existing methods are not able to reliably detect anomalous events in the resident’s activities in the presence of a visitor and to identify the time of visits, therefore generating a high false-alarm rate.

In many applications, entropy measures are used to detect the irregularities and the degree of randomness in data [16]. The research assumption is that the level of changes in a resident’s activity patterns in a home environment is an indicator of normal or abnormal activities. Hence, when the entropy value for a day exceeds the threshold value, then this indicates that there is an abnormality in the resident’s activity on that day. Distinguishing and detecting anomalies in older adults’ activities and identifying visitors (the time of their visits) is very important for healthcare management, and helps caregivers to act early to avert prospective problems. On the other hand, identifying the visit time for older adults may have a significant impact on implementing preventive social distance measures to reduce the transmission of infectious diseases, e.g., Covid-19 virus. Therefore, it is essential to develop an appropriate method or algorithm that can efficiently detect such anomalies.

Several research studies have been carried out on detecting anomalous behaviour during daily activities, using different types of sensors. These studies can be classified into three main categories, namely; ambient sensor-based [17], wearable sensor-based [18], and camera vision-based methods [19]. Ambient sensors can be easily installed in the home environment and allow people to continue with their daily routines without feeling restrained by technology. However, the use of wearable sensors and camera vision-based methods are not widely accepted by many users, especially for the older adults, mainly because they may forget to wear these sensors, may take them off when they become uncomfortable, or they are concerned about their privacy [15]. This research aims to investigate whether entropy measures can be utilised to detect and distinguish anomalies in ADLs in the presence of a visitor, and specifically in a sleeping routine and in identifying visiting times, solely based on information gathered from low-cost, non-intrusive ambient sensors.

The proposed entropy measures have been applied in two repetitions. In the first iteration, they are used to reveal days with abnormal behaviours, leading to the detection of days on which an abnormality occurred. In the second iteration, they are utilised to detect days with anomalies as well as to identify the potential causes of an anomaly by computing entropy measures. The distinction between normal and abnormal entropy values is achieved by finding the maximum entropy value on normal days as a threshold to detect anomalies in an ADL. When the entropy values exceed the threshold, then this indicates an anomaly in ADLs. This means that by finding the maximum entropy value on normal days of ADLs, it is possible to detect abnormal behaviours in human ADLs in completely unseen data.

The rest of this paper is organised in the following order. Related studies concerning anomaly detection in ADLs and identifying visitors in a home environment are presented in Section 2. In Section 3, the proposed method for exploring abnormality in a sleeping pattern and identifying the time of visits by visitors in a home environment based on different entropy measures is presented. Applied entropy measures are described in Section 4, followed by a description of the dataset employed for validation of the proposed method, the experimental results, and a robust analysis in Section 5. Finally, the pertinent conclusions of this paper are drawn in Section 6.

## 2. Related Work

In this section, a review of related work to detecting anomalies in activities of daily living and identifying visitors representing a multi-occupant environment is presented.

### 2.1. Anomaly Detection in ADLs

Anomaly detection in daily activities is a challenging task, as it depends on a specific context and the unconstrained variability of practical scenarios. Several research studies have been carried out on the detection of various types of anomalies utilising different computational methodologies, including a Hidden Markov Model (HMM) [20,21], Recurrent Neural Networks (RNN) [22], Convolutional Neural Network (CNN) [9,12], and Random Forest (RF) [23] approaches. The researchers in [20] have used HMM for anomaly detection in the daily activities sequence. Their experiments were based on data generated synthetically from a real-world dataset. Authors have shown that their proposed model can detect anomalies in ADLs with an accuracy of 95.10%. Similarly, in [21], the authors proposed an anomaly detection approach based on a dynamic Markov model. The performance of their proposed anomaly detection approach was based on both synthetic and real-world data. Their research aimed to address the challenge of reducing false alarms compared to existing techniques. The experimental results obtained from this work indicated that the proposed approach achieved the highest true positive rate and lowest false alarm rate compared to other methods mentioned in their literature review. Nevertheless, in both [20,21], the authors have provided few details about the usage of synthetic data and how the work was conducted.

A relatively new research work proposed in [11] is a novel method, positive-unlabelled deep metric learning method for anomaly detection (PUMAD), which effectively identifies various anomalies. They tested and evaluated their proposed method based on two datasets. Their results show that PUMAD has achieved good performance. However, the authors also state that the PUMAD method has some limitations in terms of its potential unsuitability for normal data that has a lot of classes (clusters). It is also reported that some further research is required to improve the proposed method by extending the study to a more generalised positive unlabelled anomaly setting, as a multi-class anomaly detection setting. In [9], a combination of CNN and Long Short Term Memory (LSTM) is used for detecting simulated anomalies in ADL data related to dementia in smart homes. The experimental results show that CNN with LSTM achieved an accuracy of 89.72%. However, the proposed method could not identify every type of anomaly in daily activities. Similarly, in [24], LSTM has been applied for anomaly detection in a sequence of daily activities in a home environment. The study aimed to compare the performance of LSTM with HMM to detect anomalies in ADLs under various sizes of training sets. The authors used the “Aruba” dataset, publicly available from the CASAS dataset [25], to test and evaluate their proposed model. The experimental results reported that both LSTM and HMM achieved the same accuracy of 87.50%; nevertheless, the proposed model could not detect all anomalies in daily activities.

Recently, several other techniques have been used to distinguish between normal and abnormal cases in ADLs. A novel version of the Gated Recurrent Unit (GRU), called Single-Tunnelled GRU, was proposed in [19] for anomaly detection and generalisation in videos. They trained and tested their proposed model based on three well-known video anomaly detection datasets. The researchers indicated that their proposed model achieved better performance than standard recurrent networks. However, the proposed model required some further work in order to improve the introduced model by fusing it with other variants of recurrent and deep networks. Novelty detection algorithms have also been applied for anomaly detection in ADLs and other datasets. Authors in [26,27] have used a One-Class Support Vector Machine (OC-SVM) to detect anomalies in daily activities. Likewise, OC-SVM has been applied for online personal risk detection based on behavioural and physiological patterns [13] and detecting anomalies in time series data [28].

The majority of the proposed anomaly detection methods in daily activities are too simplistic and thus generate a high false alarm rate. An approach with a high rate of false-alarms might not be appropriate to reliably detect anomalies in ADLs, which leads to dissatisfaction on the part of users and caregivers [29]. In order to restrict the false-alarm rate, human behaviours need to be recognised and monitored accurately. This can be achieved by using an appropriate technique, such as an entropy measure, which enables analysis to distinguish between normal and anomalous cases in daily activities with a high degree of accuracy.

### 2.2. Visitor Identification

Distinguishing and detecting a visitor in a single-occupancy home environment (represented as a multi-occupancy environment) based on ambient sensors is still a significant challenge for researchers. In recent years, some research works have been conducted on detecting and distinguishing activities in a multi-occupancy environment [4,30,31,32,33]. However, a few works have focused on visitor detection in a home environment based on ambient sensors, and especially on those with binary sensors. An unsupervised method based on a Markov Modulated Multidimensional non-homogeneous Poisson Process (M3P2) has been proposed in [34] to detect visits as anomalous activity in home environments for older adults. The research aimed to model and recognise daily and weekly activities, as well as to discriminate between regular and irregular visits in a home environment. The experimental results show that the proposed method performs better than the Markov Modulated Poisson Process (MMPP), with a total precision of 64%. Likewise, the authors in [35] proposed a new method based on MMPP for modeling regular activity patterns and detecting visits in older adults living alone in a home environment. The authors tested and evaluated the proposed method based on data collected from two apartments utilising various sensor networks. The results obtained from this study show that the proposed method for visit detection in a home environment achieved a precision of 84.2%.

The researchers in [36,37] investigated different methods in which multi-occupancy can be identified in a home environment with various sensor networks utilising an HMM and a Naive Bayes Classifier (NBC). Their experiments were tested and evaluated based on the data gathered from a video camera and binary sensors in a living lab. While these methods demonstrate a promising result, there are some constraints to the study; however, since the gathered data was limited to only one room and the number of sensors utilised was small. A Support Vector Machine (SVM) was used in [15] to detect visitors in the home environment based on data gathered from wearable devices and an ambient sensor network. The researchers indicated that their proposed method can correctly detect 58–83% of visits in a home environment. Some challenges are evident, however, such as the fact that their dataset was not fully annotated to label every visit in the home environment. Likewise, the authors in [38] applied SVM to detect periods where visitors are present in the home environment based on data collected from only motion sensors in a living lab. The number of transitions between the main living places and the number of sensor firings is extracted from the raw data and used as features in the SVM. The main drawback of the study is that visits were not recorded overnight.

Several other techniques are used for identifying activities in a multi-occupancy environment. A new research work [39] has introduced a hybrid mechanism between ontology-based and unsupervised machine learning for detecting and separating the activities of a single person in a multi-occupancy environment. They have tested and evaluated their method based on the CASAS Spring dataset. The results obtained from this work show that the proposed method achieved an average activity recognition rate of 95.83% in the context of a multi-occupancy home environment. Another new research study [4] has presented a daily activity recognition method based on time clustering for multi-occupancy in a smart home environment. The required features are extracted from the raw data using a de-noising method. Then, cluster techniques are utilised to separate activities that occur at the same location but at various times. Finally, a similarity matching method is used to complete daily activity recognition. The authors tested the performance of the proposed method based on two multi-occupancy datasets provided by the CASAS repository. The results obtained from their research indicate that the proposed method for recognition of daily activities of multi-occupancy in a smart home environment achieved an accuracy of 92%. Some other research studies have addressed the challenge of detecting daily activities in a multi-occupancy home environment using wearable sensors [40,41] or video sensors [42,43]. The major drawback of using these types of sensors is that they are not widely accepted by individuals due to privacy and ethical concerns [18,44,45,46]. Thus, it is often a preferred solution to utilise ambient sensors to identify and recognise multi-occupancy in a home environment [47,48].

Entropy measure analysis has not been given much attention and, to the best of our knowledge, none of the studies in the literature has applied any type of entropy measures for anomaly detection in daily activities in the presence of a visitor. Details of why entropy measures can be considered as an appropriate alternative method are presented in the next section.

## 3. Methodology

This paper proposes a novel entropy-based method to detect anomalies in ADLs in the presence of a visitor, solely based on information gathered from low-cost, non-intrusive ambient sensors, which include Passive Infra-Red (PIR) sensors and a door entry sensor. Since the normal daily activity patterns of the resident are expected to be different when there is a visitor in the same environment or when there are conditions that affect normal behaviour, such as disrupted a sleeping pattern. The aim is to collect the ADL data from ambient sensors to detect the anomalies in ADLs (here, identifying visiting times and irregular sleep). The challenge addressed in this paper is to avert the need to utilise a camera vision-based approach or wearable sensor to detect the anomalies in a resident’s activities, and also to identify visiting times when there is a visitor.

The research hypothesis is that the level of changes in the occupant’s activity patterns in a home environment is an indicator of normal or abnormal behaviours in ADLs. Therefore, the proposed entropy measures are based on finding the maximum entropy value in normal daily activities, which will be used as a threshold to detect abnormal behaviours in ADLs in completely unseen data. This means that any value that surpasses the computed maximum value for entropy on normal days will be indicated as an anomaly behaviour in the ADLs. Furthermore, the entropy measures are not only used to detect anomalies in ADLs, but also to identify the potential causes of anomalies. This is achieved by distinguishing whether the anomaly was the result of abnormal behaviour (e.g., sleeping disorder) or when there is a visitor to the same environment which naturally disturbs the normal activity.

A schematic diagram of the proposed entropy measures for anomaly detection in ADL in the presence of a visitor is illustrated in Figure 1, which consists of four processing stages.

In the first stage, sensor data representing ADLs in a home environment is gathered based on PIR motion detectors and door entry sensors and then pre-processed. The required numerical features to be used for computing the input vector sequences of the entropy measures are extracted from the raw data. The values of this vector are then utilised as an input vector for entropy measures. The example provided in Section 5.1.2 will elaborate on the details about the process of how the required numerical features are obtained from the dataset.In the second stage, the entropy measures are applied to the extracted vector sequence from the raw data and are calculated every day. Then, the threshold is selected as the maximum entropy value of normal days to be used for detecting any anomalous days.In the third stage, the entropy measures for each of the anomalous days are computed again every hour to examine the possible causes of the detected anomalous days and to identify any hour in which an anomaly has occurred.In the fourth stage, the main door entry sensor along with entropy measures is used to distinguish between the irregular pattern in the resident’s activity and visitors. The door entry sensor is also utilised to confirm the time of visits in a home environment and, in particular, for identifying exact visiting times.

In the next section, the description of different applied entropy measures is presented.

## 4. Applied Entropy Measures

Entropy has emerged as a suitable complexity measure for the amount of disorder or uncertainty in a system or time-series data [49]. The concept of entropy is utilised in many fields of science, including statistical mechanics, information theory, neural networks, taxonomy, and mathematical linguistics [16]. Considering different methods, entropy can be utilised as a measure of randomness or uncertainty in a system. Entropy increases as the system’s randomness increases. For example, if the degree of randomness is low, the system will be organised. The ideal system is when everything is completely organised, and the value of entropy is zero. Whereas high disorder in the data will give higher entropy values, as shown in Figure 2.

In [50], Shannon proposed entropy for information theory to describe the distribution of signal components. A statistical definition of entropy can be defined as:(1)$$SE=-k\sum_{i}p\left(i\right)\ln p\left(i\right),$$
where p(i) is the probability that it occurs during the system’s fluctuations and *k* is Boltzmann constant.

Numerous entropy algorithms have been proposed and are extensively utilised to quantify the irregularity of signals and image-processing applications [51]. The computations, however, are frequently confronted with the challenge of an insufficient number of data points. Moreover, certain recorded data are, to a certain degree, contaminated by noise. To deal with this problem of short and noisy recordings in physiological signals, Approximate Entropy (ApEn) was proposed in [52] to avert challenges in the finite length of a time series and in the need to discriminate the nature of the generating systems. High regularity and low randomness in the data produce smaller entropy values, whereas less regularity gives higher entropy values. However, the disadvantage of ApEn is that it lacks relative consistency and, furthermore, it is strongly dependent on the length of a time series [16]. Authors in [53] introduced Sample Entropy (SampEn) to overcome the drawbacks of ApEn by excluding self-matches; thus, decreasing the calculation time by one-half in comparison with ApEn. The SampEn is less dependent on the data length and shows relative consistency; however, matching vectors in both ApEn and SampEn are either 1 or 0 values. Therefore, this is not realistic when dealing with real-world examples where boundaries are not fixed [54]. To overcome such cases, Fuzzy Entropy (FuzzyEn) was proposed in [55] as a method to compute the regularity in a time series. In FuzzyEn, the concept of an exponential function, exp$\left(-{\left(d_{ij}^{m}\right)}^{n}/r\right)$, is applied as a fuzzy function that evaluates the similarity degree of two points (vectors).

A further commonly utilised regularity indicator is Permutation Entropy (PerEn), proposed in [56]. It is based on the arrangement relations between signal values and the measure of the relative frequencies of ordinal patterns. The PerEn is considered a simple measure that generates fast calculations. However, the measure does not consider the variation among amplitude values and the average value of amplitudes [51]. Existing entropy measures, such as ApEn, SampEn, PerEn, and FuzzyEn are widely utilised to measure the irregularity of signals at single-scale. Nevertheless, these measures fail to compute the multiple time scales engrained in biomedical recordings [57]. To overcome this limitation, Multiscale Entropy (MSE) was proposed in [58] and it is employed to quantify the irregularity of univariate time series, notably physiological time series.

The possibility of using entropy to determine the degree of disorder or uncertainty in a system resulted in the definition of different types of entropy. Figure 3 shows various forms of entropy measures presented in chronological order. Entropy measures have been utilised to solve essential problems in time-series analysis, such as detecting missing points in time-series, predicting appearance events, and determining similarities between time-series [16]. Entropy analysis is an established method for irregularity detection in many applications; however, it has rarely been applied in the context of ADL and Human Activity Recognition (HAR).

To evaluate the relevance of entropy measures to detect anomalies in ADLs, the following entropy measures are investigated,

-Shannon Entropy (ShEn) [50]-Approximate Entropy (ApEn) [52,54]-Sample Entropy (SampEn) [53]-Permutation Entropy (PerEn) [56]-Multiscale-Permutation Entropy (MPE) [59,60]-Fuzzy Entropy (FuzzyEn) [55,61]-Multiscale-Fuzzy Entropy (MFE) [62,63]

Readers are referred to cited paper including our earlier publication [47] for a detailed explanation of the entropy measures. These measures are applied to the same data type, and their performances are compared.

## 5. Experimental Setup

The dataset utilised for the validation of the proposed method is described in this section followed by the application of entropy measures, as used for anomaly detection in ADL in the presence of a visitor. Finally, the obtained results and performance evaluation of the proposed methods are also described.

### 5.1. Data Description

A dataset representing the ADL of a single resident is employed for the validation of the proposed methodology for identifying anomalies in ADL in the presence of a visitor. Further details about the dataset and pre-processing are provided below, followed by the experimental results in the next section.

#### 5.1.1. Activities of Daily Living Dataset

The dataset was gathered by our research team from a real home environment representing the ADL of a single resident for a period of 65 days. The dataset was collected at the SmartNTU home facilities within Nottingham Trent University. The house is equipped with several low-cost, non-intrusive ambient sensors such as PIR sensors, a pressure sensor on the sofa and bed, and door entry sensors, which are utilised as data collection devices. The PIR sensors are commonly used to track the movement of an occupant representing the occupancy of a specific area at home. They measure infrared light radiating from objects in its field of view. Hence, they can sense motion, and they are used to detect whether a human (or pet animal) has moved in or out of the sensors range. It is essential to place the PIR sensors in the right location to capture and monitor the occupant’s movements in different areas. Whereas the door entry sensors are used to detect the open and close status of the door. Due to privacy, cost issues, and ethical concerns, these sensors are the most widely used for ADL monitoring, as they allow individuals to live normally without feeling restrained by the technology [47]. Moreover, these sensors track the resident’s interaction in different locations in the house. A floor plan of the house and sensor locations utilised for data gathering are shown in Figure 4. The data gathered by these sensors are binary in the form of 1 and 0 signifying active and inactive states, respectively. In total, the dataset contains 56 normal days of ADLs, including, sleeping, eating (dining room activity), toileting, and going out of the home, etc., and 9 abnormal days of ADLs. In addition, the abnormal days contain different abnormalities in the resident’s activity, such as irregular sleep and the presence of a visitor on some days. Besides this, the information that can be obtained from the dataset is the date, start time, end time, and the location of activities, as shown in Table 1.

Consideration of ethical issues prior to data collection is an important step to protect the rights of participants and inform them about the procedures. The data collection for the above experiment was conducted using a research team member, and the research was conducted according to the institutional ethical approval process.

#### 5.1.2. Data Pre-Processing

The datasets described above include different ADLs. For this work, only PIR sensors representing the resident in an area of the house and door sensor are selected and utilised. The relevant features that can distinguish between normal and anomalous cases in daily activities are selected. The required numerical features representing ADLs from the sensor data are:Start time: this is the starting hour and minutes of entering each location (room) in the home.Duration: this is the duration in minutes the resident spent in each room, which is obtained by subtracting the end time from the start time.The transition between the rooms: this is the transition from the location of the performed activity to another location inside the home.Encoded daily activities sequence: This is the collection of activated sensor locations at different times, in which each location (room) is encoded by replacing each activity and/or location of the performed activity with an odd number (e.g., toilet = 1, bedroom-sleeping = 3, corridor = 5, kitchen = 9, etc.).

The input of any entropy measure should be formulated as a vector sequence (time series). Thus, to make the dataset appropriate for entropy measures, the encoded dataset is converted to a set of data points equally spaced in time, which is dependent on the computational time of the entropy measure. The encoded daily activity sequence is then utilised as an input vector for entropy measures.

To clarify the process of how the vector sequence is obtained from the dataset, a step-by-step example is provided below. Consider the activity data sample presented in Table 2. Firstly, the required numerical features to be used for calculating the vector sequences are extracted from the raw dataset. Then, the daily activity sequence is encoded by replacing each location (room) with an odd number, as shown in the fourth column of the Table. Finally, the features extracted from the raw data are used with the encoded daily activities as input vector sequences to the entropy measures. The entropy measures are computed every hour, which means that there are 60 samples per hour. For the sample data presented in Table 2 from 12:00 to 13:00, so the activity sequence vector $A_{N}$, which consists of 60 samples of the encoded daily activity equally spaced in time, is then defined as:$$A_{N}=\left[\underset{\mathrm{Duration}}{\underbrace{3,3,3,3,3}},5,1,1,1,1,5,7,7,7,7,7,7,7,7,...,7\right]$$

It is obvious from the given vector sequence $A_{N}$ that the time spent in each room (duration) is represented as a repetition of the same number. This process will be repeated every hour to compute the values of entropy measures.

### 5.2. Entropy Measures for Anomaly Detection

The entropy measures mentioned earlier are applied to the encoded data vector sequence to measure normal/abnormal patterns and detect anomalies in ADLs, and specifically in an irregular sleeping routine and identifying visiting times. The entropy measures are computed every day at 60 samples per hour (60*24) to identify anomalous days. This means that the vector sequence, $A_{N}$, consists of 1440 equally spaced samples. The vector sequence, $A_{N}$, is used as the input vector for the entropy measures to reveal days with abnormal behaviours, leading to the detection of days on which an abnormality occurred. To compute the ApEn, SampEn, FuzzyEn, and MFE, the parameters of embedded dimension, *m*, and tolerance, *r*, are required to be defined. Thus, the algorithm for these entropy measures is impacted by the selection of these values. The best results were obtained when the values of the parameters *m*, and *r* are 2, and 1 respectively. Whereas the values of the parameters *m* and time delay τ, which are required to compute PerEn and MPE are set as 2 and 1, respectively. After the entropy measures have been computed, the threshold is selected as the maximum entropy value of normal days to be used for detecting anomalous days. When the entropy value of each day goes beyond the calculated maximum value for entropy on normal days, it is treated as anomalous days in the resident’s activity.

It is possible to compute entropy measures at different time scales (e.g., 15, 30, 60, or 120 min) to identify anomalous days in the resident’s activity. However, Our earlier work has concluded that when the calculation period of entropy measures is less than one hour, it is not sufficiently reliable enough to detect anomalies in ADLs. This can be justified by the fact that decreasing the computational period of entropy measures will reduce the number of observations per time period, which will increase the variance. Consequently, the number of false positives will increase, which reduces precision. Therefore, the best performance is obtained when the computational time of entropy measures is performed based on a one-hour time period.

The proposed method is based on the assumption that when the entropy value of each day exceeds the threshold value, then this indicates that there is an abnormality in the resident’s activity on these days. Thereby, the proposed method can detect anomalous behaviour in unseen human abnormality data, which means the proposed method is capable of adapting to detect abnormal behaviour in ADLs in completely unseen data. Figure 5 and Figure 6 show the results obtained from applying the ShEn and FuzzyEn measures for identifying any anomaly in ADLs in the presence of a visitor, respectively. The results in Figure 5 shows that the proposed ShEn method identifies only 7 days (days 16, 29, 33, 38, 49, 52, and 63) as anomalous days in the resident’s activity out of 9 anomalous days, and failed to detect 2 of the anomalous days (days 25 and 42). This can be justified by the fact that ShEn is strongly dependent on the length of the time series and in need to discriminate the nature of the generating systems [52]. However, from Figure 6, it can be seen that the proposed FuzzyEn method detects 9 days as anomalous days (days 16, 25, 29, 33, 38, 42, 49, 52, and 63) as the FuzzyEn values of these days overrode the threshold. This means that the proposed FuzzyEn method successfully identified all anomalous days in the resident’s activity, based on the gathered ADL dataset.

After detecting nine anomalous days in the resident’s activity, the entropy measures for each of these days have calculated again every hour at 60 samples per hour to examine the possible causes of the detected anomalous days and identify any hour that the anomaly had occurred. This means that the input vector sequence to entropy measures, $A_{N}$, consists of a 60 equally spaced samples. Besides, the threshold is selected as the maximum entropy value of normal days, which is also calculated again every hour, to be used for detecting any hour the anomaly has occurred on anomalous days. Therefore, when the entropy values of each hour on a given day go beyond this threshold value, this then indicates an anomaly in ADLs at that hour. This means that by finding the maximum entropy value in normal daily activities, it is possible to detect abnormal behaviour in ADLs in completely unseen data. The results in Figure 7 and Figure 8 indicate the hour that the anomaly has occurred in the detected anomalous days by applying ShEn and FuzzyEn for the gathered ADL dataset based on one-hour time periods. The threshold value for this experiment is chosen by calculating the maximum entropy value on normal days based on one-hour time periods. To detect the hour in which the anomaly has occurred in the detected anomalous days, ShEn and FuzzyEn values for each hour in Figure 7 and Figure 8 were compared the threshold value to show when entropy value has passed the threshold value. From Figure 7, it can be observed that the proposed ShEn method identified only one hour in each day of five detected anomalous days out of 13 h in nine identified anomalous days, and failed to detect any hours the anomaly had occurred on four anomalous days, which are days 25, 29, 38, and 49. Nevertheless, all anomaly hours in ADLs are correctly detected in all identified anomalous days by applying the FuzzyEn method to the ADL dataset because the FuzzyEn values for these days exceed the threshold, as shown in Figure 8.

To identify potential causes of the hours in which the anomaly has occurred in the detected anomalous days, the main door entry sensor is used to distinguish between the entropy changes caused by irregular sleep in the resident’s activity and a visitor. The finer-grained analysis provided in Section 5.4 will elaborate on the details of identifying potential causes of the hours that the anomaly has occurred in the detected anomalous days.

### 5.3. Performance Evaluation

The proposed method is based on the hypothesis that the values of entropy measures are higher than a threshold value when there are anomalies in a resident’s activity. Therefore, to evaluate the performance of the proposed ShEn, ApEn, SampEn, PerEn, MPE, FuzzyEn, and MFE measures, first, an ADL dataset representing the ADLs of a single user are manually classified as normal or as abnormal in the resident’s activity based on periods of one hour. As can be seen from Table 3, there are 203 events indicated as a normal activity of the resident and 13 events are fixed as indicating abnormalities in the resident’s activity. The first row indicates that the proposed ShEn, ApEn, SampEn, FuzzyEn, and MFE measures successfully detected all normal activity included in the ADL dataset. However, both the proposed PerEn and MPE measures identified 202 events as being normal activity out of 203 events and miss-classified only one event. The second row demonstrates that the proposed PerEn, MPE, FuzzyEn, and MFE measures successfully identified all anomalous events, while the proposed ShEn, ApEn, and SampEn measures detected only 5, 10, and 8 anomaly events out of 13 anomaly events, respectively. Based on the results shown in Table 3, it can be argued that the proposed FuzzyEn and MFE measures correctly identified all normal and anomalous events in ADLs and outperformed other entropy measures.

The performance evaluation of the proposed entropy measures for anomaly detection in ADL in the presence of a visitor is measured automatically using a confusion matrix. There are four possible results for testing anomaly detection in ADLs in the presence of a visitor, which are defined as follows:True Positive (TP): a dataset contains an anomaly in ADLs, and this is correctly detected as an anomaly in a resident’s activity.False Positive (FP): a dataset does not include an anomaly in ADLs but is incorrectly detected as an anomaly in a resident’s activity.True Negative (TN): a dataset does not contain an anomaly in ADLs and is correctly detected as normal in a resident’s activity.False Negative (FN): a dataset includes an anomaly in ADLs but is incorrectly detected as normal in a resident’s activity.

The performance evaluation is calculated for each proposed entropy measure utilising:(2)$$Sensitivity=\frac{TP}{TP+FN}$$
(3)$$Specificity=\frac{TN}{TN+FP}$$
(4)$$False\;Positive\;Rate\;\left(FPR\right)=\frac{FP}{FP+TN}$$
(5)$$False\;Negative\;Rate\;\left(FNR\right)=\frac{FN}{FN+TP}$$
(6)$$Accuracy=\frac{TP+TN}{TP+TN+FP+FN}$$

The results presented in Table 4 represents the performance of the proposed entropy measures for anomaly detection in ADLs in the presence of the visitor when they are computed over a one-hour time period. The results of specificity indicate that all the proposed entropy measures achieve a high specificity of 100%, which means that all normal daily activities are correctly detected as normal in a resident’s activity. In contrast, the results related to sensitivity show that the proposed PerEn, MPE, FuzzyEn, and MFE measures perform better than the ShEn, ApEn, and SampEn measures since they indicate a perfect sensitivity of 100%, which means that all anomalous events in ADLs have been correctly identified. Besides, the proposed ShEn, ApEn, and SampEn measures achieve a detection rate of 38.4.5%, 69.2%, and 61.5%, respectively, which means that they have a 61.6%, 30.8%, and 38.5%, respectively false-negative rate for identifying anomalies in ADLs. However, the proposed PerEn, MPE, FuzzyEn, and MFE measures show high detection rates of 100%, which means that the false-negative rate of anomaly detection in ADLs is 0%. Based on the results achieved, the proposed PerEn, MPE, FuzzyEn and MFE measures are better indices than the proposed ShEn, ApEn, and SampEn measures to detect anomalies (here, detecting visitor and irregular sleep) in behaviour when the sample data mostly represents normal activities. This also confirms that the proposed entropy measure could be used for anomaly detection in ADLs in the presence of a visitor.

### 5.4. Robust Analysis

In this work, the entropy measures are used not only to detect anomalies in ADLs, but also to identify potential causes of anomalies by calculating entropy measures on an hourly basis and then by distinguishing between irregular sleep in the resident’s activity and visitors. The distinction between irregular sleep and visitors was achieved by using the main door sensor along with entropy measures. Moreover, The door entry sensor is used along with entropy measures to confirm the present time of the visitor in the home environment. As the visitors enter and exit the home through the main door, the door sensor is used to confirm the time of visits. This will increase the performance evaluation of the proposed entropy measures. In general, door opening or closing does not necessarily mean that a visitor is present in the home environment, as the door might be opened by the main occupant; e.g., in response to a postman or a neighbor. Therefore, the presence of visitors cannot be identified only by utilising the main door sensor. Thus, entropy measures are utilised to detect anomalies in ADLs in the presence of the visitor, and then the door sensor is utilised to confirm the time of the visit.

Figure 9 shows the distinction between irregular sleep in the resident’s activity and the visitor using a door entry sensor with entropy measures. As the best results are obtained when the computational time of entropy measures is performed based on one-hour intervals, the door entry sensor is used to confirm the visiting time on each day. As can be seen in Figure 9a, the door was opened six times on day 16, but the visitor came once on that day, at 09:03 am, and stayed in the house until 09:50 am. This means that the main resident might have caused the other door events. On this day, it can be confirmed from the door sensor data that the type of anomalies included only the visitor because the entropy value on this day exceeds the threshold value when the door is opened. Whilst Figure 9b shows that the door was opened four times on day 49, the entropy values were higher than the threshold value when no door is opened or closed. This means that the resident has an irregular sleeping pattern, and this is confirmed by the time (03:00 am). Meanwhile, Figure 9c shows that the door was opened four times on day 63, but the entropy values were higher than the threshold value in three different positions. On this day, it can be seen that the entropy values were higher than the threshold value when no door is opened or closed. This means that the resident has an irregular sleeping pattern at 02:00 am and 04:00 am compared to the usual days because it cannot be confirmed from the door sensor data. However, it can be confirmed from the door sensor data that the resident had a visitor, and the visitor came 12:08 pm and stayed in the home until 12:46 pm. This means that on this day (day 63), there was irregular sleep in resident’s activity and the occupant had a visitor on this day. The identified anomalous days and possible causes of these for the ADL dataset are summarised in Table 5.

In summary, the entropy measures are useful and relevant tools to detect abnormality (here, irregular sleep and a visitor) in behaviour when the sample data mostly represents normal activities. This also confirms the possibility that the entropy measures are used to distinguish between different causes of anomalies when they are used in conjunction with data gathered from a secondary sensor.

## 6. Conclusions

An entropy-based approach for anomaly detection in Activities of Daily Living in the presence of a visitor has been investigated. The entropy measures have been applied in two scenarios. The first case was to reveal days with abnormal behaviours, leading to the identification of the days in which abnormalities occurred. In the second case, the entropy measures were used to detect anomalies in ADLs and also identify potential causes of those anomalies (here, an irregular sleep pattern and detecting a visitor) by calculating entropy measures. The distinction between normal and abnormal entropy values was achieved in the second case by finding the maximum entropy value on normal days around the clock. This meant that any value that exceeded the calculated maximum value for entropy on normal days was treated as an abnormal behaviour point.

When the entropy values for each hour on a given day exceed the threshold value, entropy measures indicate an anomaly in ADLs at that hour. This means that by finding the maximum entropy value on normal days of ADLs, it is possible to detect abnormal behaviour in ADLs in completely unseen data. To distinguish between the entropy changes caused by irregular sleep in the resident’s activity and a visitor, the main door entry sensor along with entropy measures are used to confirm the visitor’s presence time in the home environment. The experimental results show that the PerEn, MPE, FuzzyEn, and MFE measures perform much better than the ShEn, ApEn, and SampEn measures to detect anomalies in behaviour when the sample data mostly represents normal activities. The PerEn, MPE, FuzzyEn, and MFE measures show high detection rates of 100%, which means that the false-negative rate of anomaly detection in ADLs is 0%. The conclusion drawn from this research is that the PerEn, MPE, FuzzyEn, and MFE measures are considerably better than ShEn, ApEn, and SampEn measures for anomaly detection in ADLs based on data gathered only from ambient sensors.

## Figures and Tables

**Figure 1 entropy-22-00845-f001:**
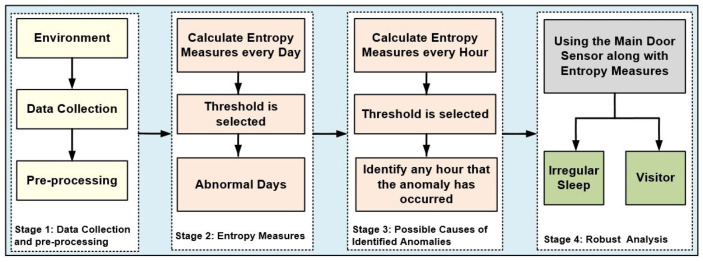
A schematic diagram of the proposed anomaly detection in activities of daily living in the presence of a visitor.

**Figure 2 entropy-22-00845-f002:**
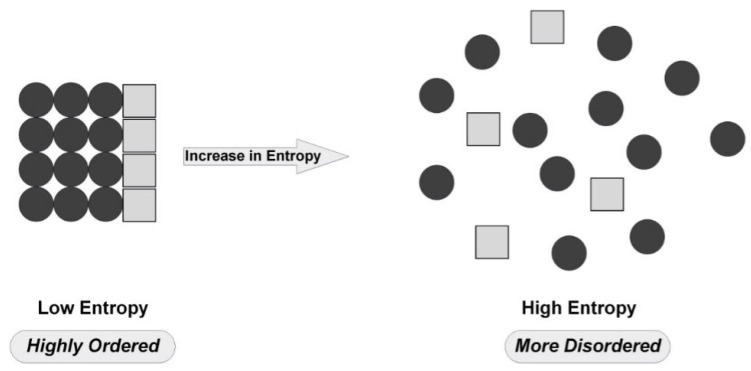
An illustration of entropy measurement definition.

**Figure 3 entropy-22-00845-f003:**
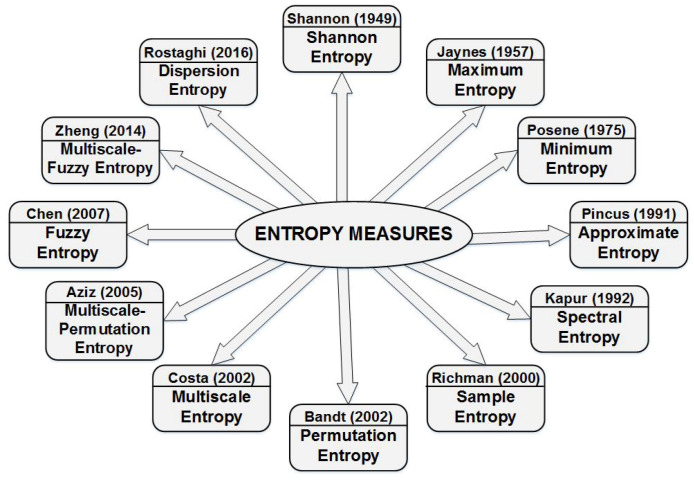
Different types of entropy measures, presented in chronological order.

**Figure 4 entropy-22-00845-f004:**
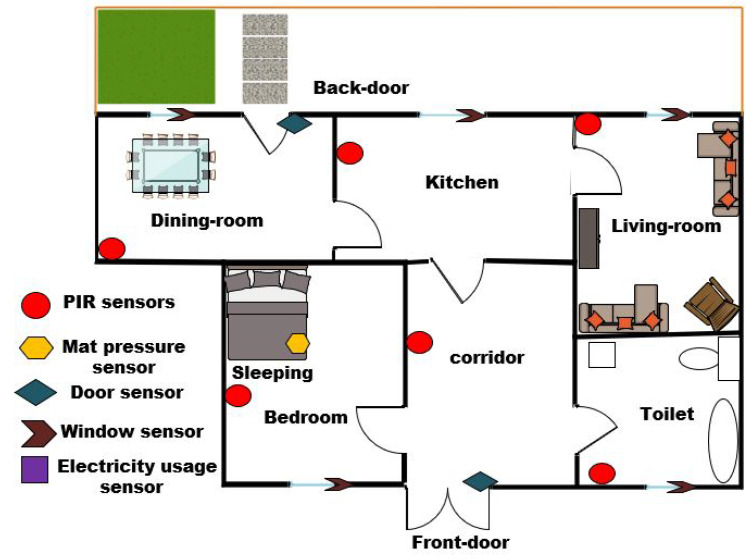
Floor plan and sensors location used for data collection in a SmartNTU home environment.

**Figure 5 entropy-22-00845-f005:**
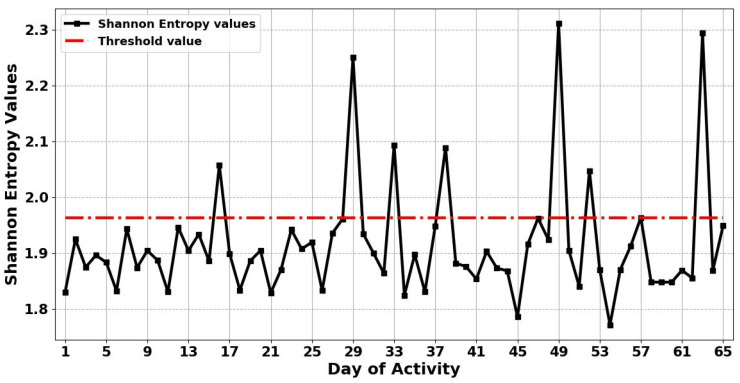
The results obtained by applying Shannon Entropy (ShEn) for anomaly detection in the activities of daily living in the presence of a visitor. The figure also illustrates the threshold value for 65 days.

**Figure 6 entropy-22-00845-f006:**
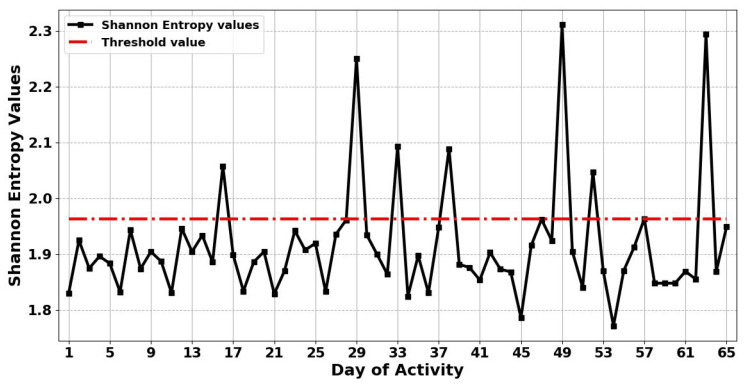
The results obtained by applying Fuzzy Entropy (FuzzyEn) for anomaly detection in the activities of daily living in the presence of a visitor. The figure also illustrates the threshold value for 65 days.

**Figure 7 entropy-22-00845-f007:**
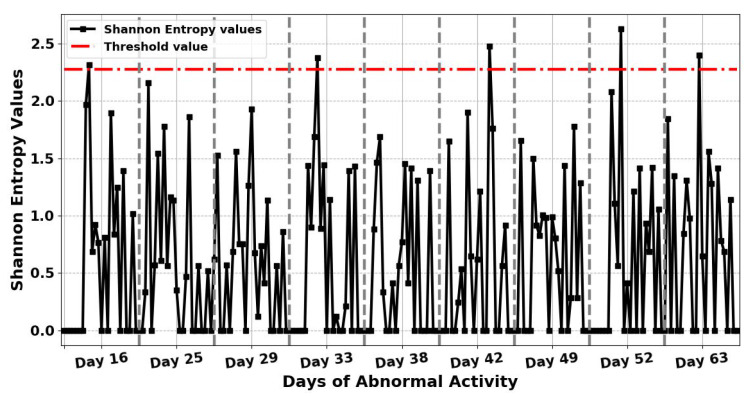
The results obtained when applying Shannon Entropy (ShEn) for the 9 days with abnormal activity to examine the possible causes of the identified anomalous days based on one-hour time periods. The figure also shows the threshold value for entropy on normal days, which will be used for detecting any hour the anomaly has occurred on anomalous days.

**Figure 8 entropy-22-00845-f008:**
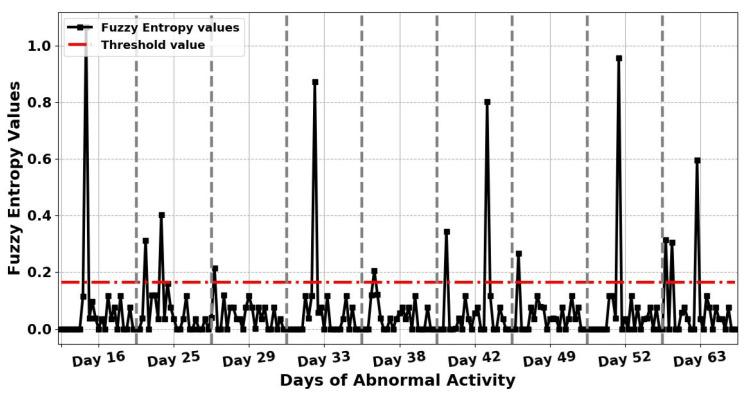
The results obtained from applying Fuzzy Entropy (FuzzyEn) for the 9 days of abnormal activity to examine the possible causes of the identified anomalous days based on one-hour time periods. The figure also shows the selected threshold value for entropy on normal days, which will be used for detecting any hour the anomaly has occurred on anomalous days.

**Figure 9 entropy-22-00845-f009:**
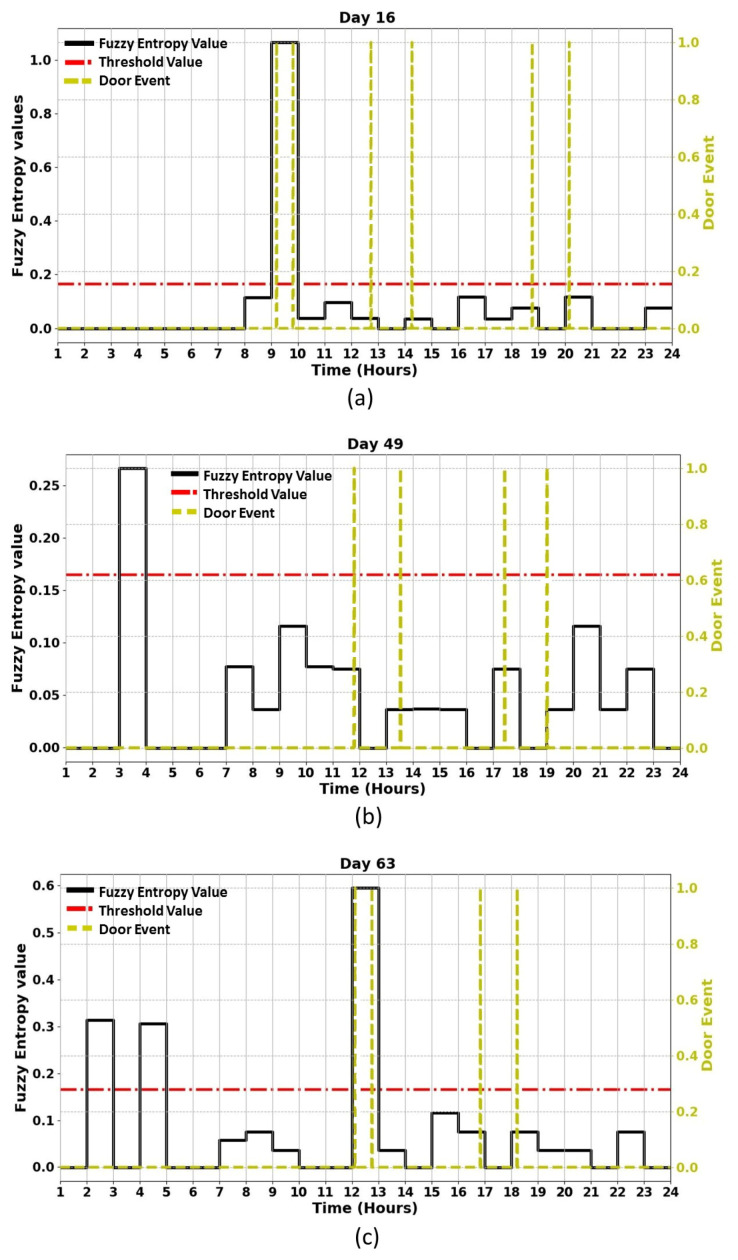
Examples of an identified visitor and irregular sleep using a door sensor with entropy measures for the collected ADL dataset representing: (**a**) visiting time on day 16, with the time confirmed using the door sensor; (**b**) irregular sleep on day 49; and, (**c**) visitor and irregular sleep on day 63.

**Table 1 entropy-22-00845-t001:** A Sample of the gathered Activities of Daily Living (ADL) dataset.

Start Time	End Time	Activity
2020-01-01 09:34:45	2020-01-01 09:38:11	Toilet
2020-01-01 09:38:26	2020-01-01 09:55:11	Bedroom-sleeping
2020-01-01 09:55:18	2020-01-01 09:55:59	Corridor
2020-01-01 09:56:07	2020-01-01 10:06:03	Kitchen
2020-01-01 10:06:14	2020-01-01 10:17:36	Dining room
2020-01-01 10:17:49	2020-01-02 11:02:44	Living room
...	...	...

**Table 2 entropy-22-00845-t002:** A sample activity data used to calculate the pre-processed input sequence vector for the entropy measures.

Start Time	Duration (min)	Location	Encoded Number of Each Location
12:00:01	5	Bedroom-sleeping	3
12:05:22	1	Corridor	5
12:06:00	4	Toilet	1
12:09:59	1	Corridor	5
12:11:00	49	Living room	7
13:00:00	6	Kitchen	9

**Table 3 entropy-22-00845-t003:** Detection accuracy of ShEn, ApEn, SampEn, PerEn, MPE, FuzzyEn, and MFE for the ADL dataset.

Events	Total Samples	Detected							Not Detected						
		ShEn	ApEn	SampEn	PerEn	MPE	FuzzyEn	MFE	ShEn	ApEn	SampEn	PerEn	MPE	FuzzyEn	MFE
Normal	203	203	203	203	202	202	203	203	0	0	0	1	1	0	0
Abnormal	13	5	10	8	13	13	13	13	8	3	5	0	0	0	0

**Table 4 entropy-22-00845-t004:** The performance results of the proposed entropy measures for the ADL dataset when computational time is performed based on one-hour time periods.

Entropy	Sensitivity	Specificity	False Positive Rate	False Negative Rate	Accuracy
ShEn	38.4%	100%	0%	61.6%	96.2%
ApEn	69.2%	100%	0%	30.8%	98.1%
SampEn	61.5%	100%	0%	38.5%	97.6%
PerEn	100%	99.5%	0.5%	0%	99.5%
MPE	100%	99.5%	0.5%	0%	99.5%
FuzzyEn	100%	100%	0%	0%	100%
MFE	100%	100%	0%	0%	100%

**Table 5 entropy-22-00845-t005:** A summary of identified anomalies days and possible causes of these for ADL dataset.

Day	Cause	Detailed Description
Day 16, 33, and 52	Visitor	The resident receives visits on these days, which might be from family members or health care workers.
Day 25, 42 and 63	Irregular sleep and Visitor	The resident has an irregular sleeping pattern and also receives a visitor on these days, and this is confirmed by using the main door sensor.
Day 29, 38, and 49	Irregular sleep	The resident has an irregular sleeping pattern compared to the usual days.

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
