# Peer review of "An Entropy-Based Approach for Anomaly Detection in Activities of Daily Living in the Presence of a Visitor"

_entropy, 2020, doi:10.3390/e22080845_

Round 1

Reviewer 1 Report

This paper presents a study about entropy to detect irregular activities of daily living (ADL)  including sleep. Shannon Entropy, Approximate entropy, sample entropy, permutation entropy, multiscale permutation entropy, fuzzy entropy, and Multiscale Fuzzy Entropy were tested on times series collected using passive infrared sensors for detecting movement and door entry sensors. The main contribution of this study is the use of a threshold of Fuzzy entropy and Multiscale Fuzzy Entropy measures as approaches for detecting irregular activities of daily living.

The study was well conducted however some minor issues should be attended:

  1. The discussion about DLS should be improved and highlight the problem and advantages of respect to approaches based on cameras.
  2. The description of the data collection procedure should be improved. It is not clear how were organized the time series considering that acquiring data from several sensors.
  3. Why the authors no considered dispersion entropy into the study taking into account the figure 3.
  4. Why did the authors not consider other sceneries into the study ?
  5. The description of the sample time of the sensors should be improved. Finally, if the sample time was one hour, please explain how were detected irregular activities in this exact moment.

Reviewer 2 Report

  1. Paper needs to be better edited. I've attached suggestions. 
  2. A finer grained analysis might be in order. In Figs 7 & 8, it would help the reader to know more about the analysis of the anomalous days that have been identified. What caused the anomalies? 
  3. Research design: The Hypothesis on p. 2 is not a hypothesis. A better hypothesis is located on p. 11. 
  4. Authors write on p. 14 that they "observed" irregular sleeping patterns, when at best they observed the data statistics that indicated an irregularity; the authors must avoid being careless with semantics.  

Reviewer 3 Report

This manuscript presented anomaly detection in activities of daily living based on entropy measures. The proposed approach identified anomalies in a multi-occupant environment with the visitors. An experimental evaluation is conducted to detect anomalies obtained from a real home environment. Experimental results are presented to demonstrate the effectiveness of employed entropy measures in detecting anomalies in the resident’s activity and identifying visiting times in the same environment. I have the following concerns for the study.

In 3. Methodology

The sensors were deployed to collect data for computing entropy. What are the criteria to define the "required numerical features" and the "anomalous days"? For example, how many visitors were measured for anomalous days in the study? How about the residents who are routinely visited by constant visitors of different groups?

In 5. Experiment setup

The study employed a single resident to detect activities of daily living. Is the participant the member of the research team or others? I think an ethical concern is requested for informed consent to whom was not a team member but was monitored in the test. The approval by the Institutional Review Board (IRB) should be provided.

Regarding the experiment design, the capability of various sensors is expected to described briefly in a sub-section for ensuring reliable data collection. For example, a PIR sensor can detect a human within a range of meters (i.e. the room space is available for detection).

The entropy measure for anomalous detection describes the parameters such as m, r, tau. Are they arbitrarily defined or with particular meaning?

Reviewer 4 Report

This work presents a current topic of high relevance whose information can be used for public health policies.

The work is presented correctly and has excellent quality; therefore, it can be accepted in the current form.

My only suggestion is to try to explain why Shannon entropy does not achieve the success rate compared to the analysis using the method proposed here. Perhaps you can delve a little into this result.

Round 2

Reviewer 1 Report

The authors attended all suggestions.

Reviewer 2 Report

I've made 2 minor recommended changes in the attached review of the revised manuscript

Reviewer 3 Report

I don't have more question. The paper is available for acceptance.